# Benchmarking Data Sets from PubChem BioAssay Data: Current Scenario and Room for Improvement

**DOI:** 10.3390/ijms21124380

**Published:** 2020-06-19

**Authors:** Viet-Khoa Tran-Nguyen, Didier Rognan

**Affiliations:** Laboratoire d’Innovation Thérapeutique, UMR7200 CNRS-Université de Strasbourg, 67400 Illkirch, France; vktrannguyen@unistra.fr

**Keywords:** PubChem BioAssay, benchmarking, data set, assay selection, false positives, chemical bias, potency bias, data curation

## Abstract

Developing realistic data sets for evaluating virtual screening methods is a task that has been tackled by the cheminformatics community for many years. Numerous artificially constructed data collections were developed, such as DUD, DUD-E, or DEKOIS. However, they all suffer from multiple drawbacks, one of which is the absence of experimental results confirming the impotence of presumably inactive molecules, leading to possible false negatives in the ligand sets. In light of this problem, the PubChem BioAssay database, an open-access repository providing the bioactivity information of compounds that were already tested on a biological target, is now a recommended source for data set construction. Nevertheless, there exist several issues with the use of such data that need to be properly addressed. In this article, an overview of benchmarking data collections built upon experimental PubChem BioAssay input is provided, along with a thorough discussion of noteworthy issues that one must consider during the design of new ligand sets from this database. The points raised in this review are expected to guide future developments in this regard, in hopes of offering better evaluation tools for novel in silico screening procedures.

## 1. Introduction

The PubChem BioAssay database (http://pubchem.ncbi.nlm.nih.gov/bioassay) was first introduced in 2004 as a part of the PubChem project initiated by the National Center for Biotechnology Information (NCBI), aiming to provide the scientific community with an open-access resource where experimental bioactivity high-throughput screening (HTS) data of chemical substances can be found [1,2,3,4,5]. Starting out with small-molecule HTS input from the National Institute of Health (NIH), the database now gathers data from over 700 different sources, including governmental organizations, world-renowned research centers, and chemical vendors, as well as other biochemical databases, featuring over 260 million bioactivity data points reported in both small-molecule assays and RNA interference reagents-screening projects [5,6,7,8,9,10,11]. Journal publishers are also acknowledged for significant contributions to the growth of PubChem BioAssay, as the database has received experimental input from more than 30 million scientific publications in response to requests from over 400 peer-reviewed journals (as of 30 April 2020) [10,11,12], denoting a constant and tremendous effort from many sectors of the scientific community to support the free sharing of HTS data.

Soon after its introduction, PubChem BioAssay has established itself as a reliable and highly queried public repository where information on each biological assay, from overall descriptions to detailed screening protocols and from input data to assay results, as well as the chemical features and bioactivities of all tested molecules, can be easily accessed and downloaded directly from the webpage. The two search options (limits search and advanced search) allow a systematic and thorough investigation of the assays deposited on the database, according to various parameters, e.g., assay type, target type, or quantity of featured substances, offering a practical data collection and analysis tool [13]. Information on related targets and same-project assays enables a more complete look into the body of screening campaigns on the same or closely related biological targets. Crosslinks to the NCBI Entrez information retrieval system [14], PubMed Central [15], and the Protein Data Bank [16] also facilitate research relying on the use of data extracted from the resource. Various updates have been brought to PubChem BioAssay over the years, enlarging the size of the available archival data, introducing new features to the web interface, and improving the data-sharing capability [17,18,19,20]. Several million users have been procuring data from the website and its different programmatic services each month [21], highlighting the importance of this public database as a key source of chemical information for researchers, students, and the general public from around the world.

In this review article, a quick summary of the assays and compounds deposited on PubChem BioAssay, along with an overview of data sets built by the cheminformatics community upon the data retrieved from this repository, will be provided. We also give a thorough discussion of noteworthy issues that have to be addressed prior to utilizing such data in cheminformatics-related projects, with illustrations observed in our recently introduced LIT-PCBA data collection [22], which was constructed from PubChem BioAssay data.

## 2. PubChem BioAssay Statistics: Assays and Compounds

As of 30 April 2020, there were 1,067,896 assays deposited on the database. The vast majority of them (99.98%) involved small-molecule screening; only 177 assays were conducted with RNA interference reagents. These assays are classified according to the number of tested substances (chemical samples provided by data contributors [8]), the number of active substances, the screening stage, and the target type, as listed in Appendix A. It can be deduced that most PubChem assays are small-scale screening projects, with over 99% of them conducted on fewer than 100 substances and nearly 94% giving no more than nine actives (Figure 1). The screening stage was, in most cases (about three-quarters), not specifically annotated. Assays giving confirmatory results regarding the bioactivities of tested molecules account for a larger proportion than primary screens, though dose-response curves are not always provided. Interestingly, nearly 75% of available assays do not have a specific biological target (i.e., a protein, a gene, or a nucleotide) but are rather cell-based assays identifying molecules that interfere with a certain cell function or an intracellular activity (e.g., tumor cell growth inhibitors, lipid storage modulators, or HIV-1 replication inhibitors) or are pharmacokinetics studies. On the other hand, some assays take multiple macromolecules as targets, e.g. AID (assay identifier) 1319. The utility of data extracted from these assays in cheminformatics-related research will be later discussed in the manuscript.

A total of 102,694,672 compounds were tested in at least one PubChem bioactivity assay (as of 30 April 2020), over 95% of which were organic molecules (i.e., molecules bearing no atom other than H, C, N, O, P, S, F, Cl, Br, and I). The term “compounds”, according to PubChem, refers to unique chemical structures that were extracted and standardized from the community-provided substances [8]. A question always raised when it comes to drug design is whether a chemical compound is drug-like or not, or if a molecule has physicochemical properties that are deemed favorable for oral administration in humans. Several rules of thumb have addressed this issue, giving criteria largely employed to predict a compound’s drug-likeness, including the Lipinski’s rule of five [23,24], the Ghose filter [25], and Veber’s rule [26]. PubChem compounds are analyzed according to each criterion [23,24,25,26,27], and the statistics are given in Appendix A. Statistical results show that most compounds tested in PubChem bioactivity assays satisfy the aforementioned rules, indicating their potential to become orally active drugs (Figure 2). However, only 1% of them (over 1 million compounds) were deemed active in at least one screening experiment, highlighting the miniature portion of active molecules available in the database and implying an average “hit rate” lower than those observed in artificially constructed data sets such as DUD [28], DUD-E [29], or DEKOIS 2.0 [30]. The other compounds were either biologically inactive in all assays where they were tested or were left “inconclusive” in terms of bioactivity. These “inconclusive” compounds, present in various AIDs such as 1345009, 1345010, or 743075, have to be discarded when data extracted from PubChem BioAssay are used in cheminformatics-related research. On the other hand, compounds being repeatedly inactive in HTS assays, dubbed “dark chemical matter” [31], are, in fact, important to keep, notably for identifying ligands of novel targets (e.g., protein-protein interfaces).

## 3. What We Can Do with PubChem BioAssay Data: From the Data Set Construction Point of View

Being a wealth of experimental bioactivity data constantly gathered from many parts of the world, PubChem BioAssay offers ample opportunities for scientists from various disciplines, e.g., biochemistry, pharmacy, or cheminformatics, to exploit this abundant resource for both teaching and researching purposes. Access to the database is facilitated by numerous online services, in both manual (via PubChem limited and advanced search engines [32,33]) and programmatic ways (via access routes such as the Power User Gateway (PUG) [34], PUG-SOAP [35], PUG-REST [36], PUG-View [37], the PubChem RDF REST interface [38], or the Entrez Utilities [14]). Recently, a novel web service called ScrubChem was introduced [39], gathering PubChem BioAssay data that were already reparsed, digitally curated, and improved, allowing a systematic analysis of all targets, chemicals, and assays featured on the database at low computational costs, after which the cleaned data can be downloaded for use in modeling applications. Upon acquiring experimental input from the resource, scientists may use it in various ways to achieve their research objectives. Several review articles have been published in this regard [7,40,41], summarizing a wide range of studies that were conducted on the basis of PubChem BioAssay data [42,43,44,45,46,47,48,49,50,51,52,53,54,55,56,57,58,59,60]. In this section, we only place our focus on the research featuring benchmarking data collections that were constructed by the cheminformatics community from PubChem’s experimental results as a means of validating in silico screening protocols.

Throughout the years, various artificially constructed data sets have been developed [28,29,30,61,62,63,64,65,66,67,68,69,70,71], including DUD, DUD-E, or DEKOIS 2.0. However, the design of these collections suffers from many drawbacks, as demonstrated in several studies [72,73,74,75,76]. One of them is the unknown potency of presumably inactive molecules, also known as “decoys”, which were usually extracted from the BIOVIA Available Chemicals Directory (ACD) [77] or the ZINC database [78]. This means there is no guarantee that the “decoys” do not exert the desired bioactivity against the protein target, due to the lack of relevant experimental evidence, and it is therefore very likely that false negatives exist among the inactive molecules. Using data from PubChem BioAssay as the input for database construction, on the other hand, helps alleviate this problem. A number of data collections of different sizes have been designed from PubChem data and introduced to the scientific community, offering better references for evaluating novel virtual screening methods. Not counting nonpublicly available data sets (e.g., the three small- and medium-sized ligand sets that we designed in 2019 to validate our new pharmacophore-based ligand-aligning procedure [79]), in this section, we only mention open-access ones, including the Maximum Unbiased Validation (MUV) data sets [80], the UCI Machine-Learning Repository [81], the BCL::ChemInfo framework by Butkiewicz et al. [82], the Lindh et al. data collection [83], and our recently introduced LIT-PCBA (Table 1) [22].

### 3.1. The MUV Data Sets

The Maximum Unbiased Validation (MUV) data sets, built by Rohrer and Baumann in 2008 and published in early 2009 [80], are among the first benchmarking sets of compounds whose bioactivities were experimentally determined and retrieved from PubChem BioAssay, which, as a result, avoids the issue regarding unknown potency values of presumably inactive molecules (“decoys”) inherent in other data sets [80]. Based upon 18 pairs of primary HTS and corresponding confirmatory dose-response experiments, whose biological targets range from kinases, G protein-coupled receptors (GPCRs), nuclear receptors to protein-protein interactions, 17 medium-sized ligand sets (15,030 compounds), each with an active-to-inactive ratio at 2 × 10^−3^, were generated, implying smaller hit rates in comparison to those of other databases [76,80]. Specifically designed to be maximally unbiased, the MUV data sets were prepared according to a workflow that removed assay artifacts, prevented artificial enrichment, and reduced “analog bias” in the composition of their ligands. A series of consecutive filters was first applied to eliminate “false positives” among the active molecules, including promiscuous aggregators, frequent hitters exerting off-target or cytotoxic effects, as well as chemicals which are likely to spoil the assay’s optical detection method. A subsequent “chemical space-embedding filter”, encoded by vectorized descriptors related to the physicochemical properties of each molecule (e.g., molecular weight and number of hydrogen bond donors/acceptors), was next employed to rule out actives that were not adequately embedded in inactive compounds, ensuring that the inactive sets did not significantly differ from the sets of actives, thus avoiding possible artificial enrichment. Finally, a refined nearest neighbor analysis was applied, based on a “nearest neighbor function” and an “empty space function”, to reduce both the level of self-similarity among the actives and the separation degree between the active and inactive molecules, selecting only 30 true actives and 15,000 true inactives that were optimal as regards the criterion of spatial randomness for each ligand set. Post-design analyses on the resulting data sets showed that (i) there existed a large number of distinct molecular scaffolds presented by the ligands (1.2 compounds/scaffold class), denoting the absence of “analog bias” and a good representation of drug-like chemical space, (ii) the correlation between the degree of data set clumping and retrospective virtual screening performance was no longer observed after the MUV design, suggesting that the final ligand sets were indeed not affected by benchmarking data set bias, and (iii) the MUV data were significantly less biased than the then-standard DUD data set, as evidenced by a lower molecular self-similarity level and a higher difficulty in distinguishing true actives from true inactives by ligand-based virtual screening simulations. The introduction of the MUV data collection therefore marked a milestone in the quest to construct realistic data sets entirely from experimental results with little design bias and applicability to evaluate both ligand-based and structure-based in silico methods, serving as an inspiration for future database development.

### 3.2. The UCI Repository

The UCI Machine-Learning Repository was introduced in 2009 [81]. On the basis of data retrieved from 12 PubChem bioactivity assays, both primary (*n* = 7) and confirmatory (*n* = 5), a total of 21 medium- and small-sized data sets (69–59,795 compounds) were generated, either by using separately primary or confirmatory screening data, or by combining results from a primary assay and its corresponding confirmatory screen. In the latter case, compounds that were deemed as active in the primary experiments but later denounced as inactive by the confirmatory readouts were all considered inactive in the combined data sets (instead of being discarded, as in the MUV collection). The active-to-inactive ratio ranged from 2 × 10^−4^ to 0.33. Each ligand set was then randomly split into a training-and-validation set (80% of the population) and an independent test set (the other 20%) for machine-learning algorithm assessments [81]. Despite being one of the earliest remarkable attempts at using experimental data from PubChem BioAssay for data set construction, the UCI database itself has several limitations. Firstly, though the author offered 21 data sets in total, only four of them, which were built by combining primary and confirmatory results, were recommended. Reasons for this lie in (i) the high portion of false positives recorded in primary experiment-based ligand sets that casts doubt on the solitary use of such data for evaluating in silico screening, (ii) the hit rates observed in the sets built upon confirmatory assays alone are too high (7–33%) to be deemed realistic, notably in comparison to those of real screening decks, and (iii) the size of some data sets is too tiny (tens of active molecules among fewer than 100 compounds) for virtual screening methods (especially ligand-based ones) to give any meaningful results. Secondly, due to the lack of high-quality biological target 3D structures for several bioassays (e.g., AIDs 456 and 1608) and insufficient information on possible binding site(s) of the molecules, the design focus of this data collection is implied to be limitedly placed on a ligand-based (machine-learning) approach evaluation. Thirdly, the issue of physicochemical bias in the composition of active and inactive molecules that may lead to artificial enrichment and an overestimation of the virtual screening performance, which had already been raised in the MUV paper [80], was not addressed throughout the development of these data sets, raising questions on the real benefits of using such data for validating novel in silico screening procedures.

### 3.3. The Butkiewicz et al. Data Collection

Another PubChem BioAssay-based data collection was introduced in 2013 by Butkiewicz et al. as a part of the cheminformatics framework BCL::ChemInfo [82]. Nine medium- and large-sized data sets (>60,000 compounds) were constructed upon collating results from relevant confirmatory screens, thus avoiding the issue of false positives commonly observed when only primary readouts are accounted. Diverse classes of protein targets are covered in the database, including three GPCRs, three ion channels, the choline transporter, the serine/threonine kinase 33, and the tyrosyl-DNA phosphodiesterase. Active-to-inactive ratios range from 5 × 10^−4^ to 7 × 10^−3^, implying small hit rates that are all lower than 0.8% (<0.1%, in most cases). Though the number of true actives is deemed sufficiently large (>170 actives for each ligand set) and the hit rates are generally low, one drawback of this database is that the problems regarding assay artifacts, analog bias, and artificial enrichment due to physicochemical differences between active and inactive molecules (which need to be properly addressed during the construction phase) were completely overlooked. These issues are even more critical when data sets intended for evaluating ligand-based virtual screening methods (which is, in fact, the design focus of this data collection) are developed. There is, hence, no guarantee that only a little chemical bias exists in the composition of these ligand sets, and it is likely that in silico screening performances could be overestimated due to such unconsidered issues.

### 3.4. The Lindh et al. Data Collection

In 2015, Lindh et al. introduced a novel data collection designed for evaluating both ligand-based and structure-based virtual screening methods [83]. A rigorous procedure of analyzing the whole PubChem BioAssay database was first carried out—after which, only assays (excluding cell-based and multiplex ones) that were performed with more than 1000 compounds (at least 20 of which were identified as active) against a single protein target that had been co-crystallized with a drug-like molecule were kept. The sole protein structure chosen to represent each target had to be of the same species as that used in the corresponding high-throughput screen, must not be bound to any DNA fragment or cofactor other than ATP (to avoid the possibility of multiple binding sites), and had the highest resolution (<3 Å), as well as the fewest missing atoms, among the available structures on the Protein Data Bank [16]. Only 19 bioassays, both primary (*n* = 7) and confirmatory (*n* = 12), related to seven protein targets were retained. Molecules having been identified as active in primary assays but not validated by confirmatory screens were all discarded from the active ligand sets. The remaining active compounds were then subject to the Hill Slope filter (which takes inspiration from the MUV database) and the pan-assay interference compounds (PAINS) filter [84,85,86,87,88,89] to eliminate potential false positives. In the end, seven medium- and large-sized data sets (>59,000 compounds) were constructed, with active-to-inactive ratios ranging from 7 × 10^−5^ to 1 × 10^−3^, indicating hit rates significantly lower than those commonly seen in other databases. It was observed that a large number of unique Bemis-Murcko scaffolds were present among the active molecules (1.4 compounds/scaffold), implying that there was little analog bias and substantial structural diversity in the active set composition. Though no direct measure was taken to reduce the artificial enrichment due to differences between the true actives and true inactives, retrospective virtual screening on the seven final data sets using the physicochemical property similarity search (1D approach) and molecular docking was carried out, suggesting that the docking performance was not based on artificial enrichment, as the 1D method gave much lower enrichment in true actives than the structure-based approach, in most cases. The Lindh et al. data collection is therefore considered the next remarkable step towards employing experimental input from PubChem BioAssay to build realistic data sets suitable for both ligand-based and structure-based in silico screening evaluations while addressing (and avoiding, to a considerable extent) most issues inherent in many other databases, including false positives, analog bias, and artificial enrichment. However, due to the unreasonably rigorous data quality filters that were applied during the construction of this data collection, the quantity of target sets offered by the authors is relatively small (only seven), and several important protein families that have been largely investigated by biochemists, e.g., GPCRs and nuclear receptors, are neglected (only two kinases were included in the database).

### 3.5. The LIT-PCBA Data Sets

Five years later, we (Tran-Nguyen et al.) developed and introduced a novel data collection entitled LIT-PCBA [22]. A rigorous systematic search was first performed on the ensemble of PubChem bioactivity assays, keeping only confirmatory screens conducted with over 10,000 substances, giving no fewer than 50 active molecules, against a single protein target having at least one crystal Protein Data Bank (PDB) structure bound to a drug-like ligand of the same phenotype as that of the confirmed actives. A total of 21 assays corresponding to 21 targets covering 11 diverse protein families, including three GPCRs, three kinases, and five nuclear hormone receptors, were retained. Contrarily to the data sets of Lindh et al., in LIT-PCBA, all relevant protein-ligand structures available on the Protein Data Bank were kept, providing 162 “templates” in total. Taking inspiration from the MUV paper, we also addressed the issues of false positives, artificial enrichment, and analog bias during the construction of the LIT-PCBA data sets. The active and inactive substances retrieved from PubChem BioAssay were subjected to a series of consecutive filters, which ruled out inorganic chemicals (bearing at least one atom other than H, C, N, O, P, S, F, Cl, Br, and I); frequent hitters; nonspecific binders; promiscuous aggregators; spoilers of optical detection methods; compounds with extreme molecular properties; and ligand preparation failures. Physicochemical differences between active and inactive substances were mitigated, as all molecular properties of the remaining ligands were kept within the same range, thus avoiding the presence of molecules that were too different from others in terms of physicochemical features. Retrospective virtual screening by ligand-based methods (2D fingerprint similarity search and 3D shape overlapping) on the resulting data collection confirmed that there was indeed little chemical bias in the composition of the ligand sets, as both approaches generally gave comparable performances to random selection. The results from molecular docking were also considered along with those of the two ligand-based approaches, leading to the selection of 15 small- to large-sized target sets (4247–362,088 molecules) that finally constituted the LIT-PCBA collection. The active-to-inactive ratios span over a relatively wide range, from 5 × 10^−5^ to 0.05, but are below 3 × 10^−3^ in most cases, implying smaller hit rates than those of many other databases. Moreover, active substances included in LIT-PCBA are generally less potent than those found in DUD-E and ChEMBL, which imposes a more difficult challenge for in silico screening. Each ligand set was then further unbiased by the asymmetric validation embedding method (AVE) [73], yielding validation and training subsets with minimized overall bias that are ready for benchmarking novel virtual screening procedures. To the best knowledge of the authors, LIT-PCBA is now the latest attempt at constructing realistic data sets from confirmatory PubChem BioAssay data, possessing numerous advantages. Firstly, a large variety of protein targets (including heavily researched ones) are featured in the collection, and all available PDB structures are accounted. This practice takes into consideration at the same time the entire chemical diversity of known target-bound ligands and the complete conformational space accessible to the investigated target. Secondly, assay artifacts and chemical bias, as well as potency bias, in the composition of ligand sets were avoided or reduced, preventing the possible overestimation of in silico screening performances. Thirdly, the eventual data-unbiasing step based on chemical space analyses offers a rational split of every existing set of molecules (instead of the random division that was previously observed in the UCI repository design). This further ensures the absence of both obvious and hidden bias in the final data sets. Lastly, thanks to the presence of at least one high-quality 3D structure with well-defined binding site(s) that represents each protein target, and the aforementioned chemically unbiased ligand set composition, the application of LIT-PCBA is thus not intended only for evaluating ligand-based or structure-based virtual screening alone but, rather, for both and, especially, for the field of machine-learning algorithm development. There exist, however, some limitations in the design of this data collection, such as the relatively high hit rates of some ligand sets (2–5%) or the number of remaining true actives for several targets that is quite small (tens of molecules) for in silico methods to give any meaningful results. The current situation, as a consequence, still leaves plenty of room for further improvements, and more data sets based on experimental bioactivity assays are encouraged to be constructed, with inspirations taken from the existing collections mentioned above, to offer more realistic sets of molecules that mimic those employed in actual high-throughput screening campaigns and to provide a better evaluation of novel virtual screening approaches.

## 4. Noteworthy Issues with Using Data from PubChem BioAssay for Constructing Benchmarking Data Sets

As demonstrated in the literature and the previous section, data retrieved from PubChem BioAssay may be used for various purposes in cheminformatics-related research, including benchmarking data set construction. Due to the availability of a wide range of assays with diverse ligand sets that the database offers, it is important to be conscious of all the issues that may arise regarding the usage of such large data [22,80,83], in terms of assay selection and data curation, to properly employ these abundant resources.

### 4.1. Assay Selection for Evaluating Virtual Screening Methods

#### 4.1.1. Assay Selection as Regards the Data Size and Hit Rates

One of the first questions that we have to face when using data from the PubChem BioAssay repository to build benchmarking data sets concerns the assay(s) that should be chosen. As mentioned earlier in the manuscript, as of 30 April 2020, there were over a million assays deposited on the database. However, only a few of them can be deemed suitable for method evaluation purposes. There are many factors that one should consider before deciding which assay(s) to use. We herewith propose, as primary conditions to filter out unsuitable assays, the selection of only small-molecule HTS assays yielding biologically active molecules. RNAi assays, on the other hand, were conducted on microRNA-like molecules comprising twenties of base pairs that violate most drug-likeness rules of thumb and are, therefore, not of great interest in small-molecule drug discovery. For the sake of having an acceptable amount of ligands in the data that may give a meaningful retrospective evaluation of in silico screening methods, we recommend that only assays with no fewer than 10 actives selected among at least 300 tested substances should be kept. Data sets including only nine or fewer actives are considered too small and would be over-challenging for virtual screening, especially for machine-learning algorithms to learn anything meaningful. On the other hand, assays conducted with fewer than 300 substances while yielding more than 10 actives give hit rates that are deemed too high in comparison to those typically observed in experimental screening decks [22], even higher than those of existing data sets such as DUD [28], DUD-E [29], or DEKOIS 2.0 [30]. There may exist, of course, assays with high hit rates that remain after this initial check (e.g., AIDs 1, 3, 720690 and 720697); however, the aforementioned conditions are proposed to demonstrate that there is only a very small portion of available PubChem assays (0.20%) whose data may be considered for evaluating virtual screening protocols (Figure 3). The ligand sets of the remaining assays need to be further examined and may be filtered to ensure that their hit rates are as close as possible to those of experimental HTS campaigns and that they are suitable for the nature of the screening method (ligand-based or structure-based).

#### 4.1.2. Assay Selection as Regards the Nature of Virtual Screening

As demonstrated in various papers, a ligand set may be appropriate for the evaluation of only ligand-based in silico approaches [81,82], or only structure-based methods [76], or sometimes both [22,80,83]. This depends on the quantity and the chemical composition of all molecules that constitute the data set and the availability and the quality of 3-dimensional structures of relevant protein targets, as well as the definition of binding site(s) in which active substances exert their bioactivity. Data sets retrieved from the PubChem BioAssay database, being no exception, have to be thoroughly examined according to the criteria mentioned above before being used to assess a certain virtual screening method. Ideally speaking, an assay whose ligands are considered for evaluating structure-based approaches needs to be conducted on a protein target whose structure has been solved at a high resolution, with no ambiguity in terms of electron density, with at least a molecule of the same phenotype (agonist, antagonist, inhibitor, etc.) as that of the active compounds. However, targets for which no crystallographic or electron-microscopic structure is deposited on the Protein Data Bank may also be considered if high-quality homology models are available. An example of this can be seen in the assay AID 588606, featuring inhibitors of the yeast efflux pump Cdr1. Though the protein target, the ABC (ATP-binding cassette) drug-resistance protein 1 of *Candida albicans* (*Ca*Cdr1p), has not yet been available in the Protein Data Bank with a known inhibitor, a homology model of this transporter was generated using the human ABCG5/G8 crystal structure as the template, and possible binding sites located in the transmembrane domain were identified and validated by means of atomic modeling and systematic mutagenesis, confirming their essential role in Cdr1p-induced multidrug resistance [90]. However, caution should be taken when one uses such artificially constructed models as the input for structure-based screening approaches. On the other hand, the presence of many nonoverlapping binding sites (orthosteric versus allosteric) in the 3D structures of protein targets (as observed in those of AIDs 1469, 624170, or 624417), either crystallographic or not, may ultimately become a reason for failures in screening PubChem molecules on such proteins, especially when there is no information on the exact binding site of the tested substances that can be deduced from the assay description [22]. As virtual screening performances may vary quite significantly depending on the protein structure employed as the input [22], one should therefore be cautious when using data of these assays for evaluating structure-based screening procedures, lest they give poorer performances than expected due to external reasons that are not related to the methods themselves. Another point that should not be overlooked concerns assays that were conducted on substances derived from only a few chemical series, as they may give rise to bias that overestimates the screening performance, notably that of ligand-based approaches. If another similar assay on the same target but with a more diverse ligand set (in terms of chemical features) is available, one is recommended to make use of this assay instead. Otherwise, the “biased” data need further tuning to be deemed suitable for evaluation purposes, e.g., by filtering out “redundant” compounds (this point will be thoroughly discussed in the next section of this manuscript). However, this ligand-filtering process should not lower the number of active substances to a value so small that ligand-based methods or machine-learning algorithms cannot come up with meaningful results.

#### 4.1.3. Assay Selection as Regards the Screening Stage

Additionally, the use of data from “primary assays” should be subject to caution, as the activity outcome was only determined at a single concentration and has not yet been validated on the basis of a dose-response relationship with multiple tested concentrations [3,91]; hence, the potency values of active molecules are not confirmed. As a matter of fact, some substances originally deemed as active in a primary assay may be denounced as inactive by a subsequent confirmatory screen, as seen in AIDs 449 and 466 or AIDs 524 and 548. We therefore recommend that primary screening data should only be used if there exists a confirmatory assay that validates the potency of the selected active molecules. This practice was already observed in the construction of the MUV data sets by Rohrer and Baumann [80], in which pairs of primary and corresponding confirmatory screens were employed, whose data were then combined to form the final ligand sets. In this manner, the large pool of inactive substances from the primary assay is not neglected, and the bioactivities of the confirmed hits are indeed guaranteed, affording a vast data set (usually implying a low hit rate) with fully validated active components. Otherwise, the output data of the primary screens alone should be used with great caution, due to the risk of assuming “false positives” that may later falsify the virtual screening outcomes. An exhaustive search on the whole PubChem BioAssay database is therefore of paramount importance to select relevant data sets for the retrospective assessment of in silico screening protocols in order to ensure the quality of such evaluations.

### 4.2. Detecting False Positives among Active Substances

Concerns have long been raised over the presence of chemical-induced artifacts in screening experiments, leading to false positive findings among the molecules deemed as active [22,80,83,84,85,86,87,88,89,92]. Misinterpretation of the assay results and subsequent inaccurate conclusions may stem from various reasons largely discussed in the literature. Among them are off-target effects of compounds exerting unspecific bioactivities, possible biological target precipitation by organic chemical aggregations, inherent fluorescent properties of substances that interfere with fluorescence emission detection methods, or luciferase inhibitory activities of molecules that spoil light emission measurements in reporter gene assays [80]. Active substances whose modes of action are subject to the aforementioned issues must therefore be removed from the PubChem BioAssay ligand sets before the data can be used for retrospective virtual screening purposes. Rohrer and Baumann (2009) addressed this problem during the construction of their MUV data sets from the database, designing a so-called “assay artifacts filter” aiming to eliminate all active ligands that likely become false positives, thus prevent them from affecting subsequent screening performances. The filter is composed of three filtering “layers”, including (i) the “Hill slope filter” after which the actives whose Hill slopes for the dose-response curves are lower than 0.5 or higher than 2 are eliminated, (ii) the “frequency of hits filter” that keeps only the molecules deemed as active in no more than 26% of the bioactivity assays in which they were tested, and (iii) the “auto-fluorescence and luciferase inhibition filter” that rules out compounds exhibiting auto-fluorescent properties along with inhibitors of luciferase [80]. All frequent hitters, unspecific binders (molecules with multiple binding sites), experimentally determined aggregators, and spoilers of optical detection methods are, as a result, removed from the PubChem data sets after these filtering steps. Such filters indeed have a profound impact on the population of active substances, as over a half of them were deleted by these “false positives filters” during the development of our recently introduced LIT-PCBA data set (Figure 4) [22]. This drastic decrease in the number of confirmed actives also helps lower the “hit rates” observed in our ligand sets (as only the actives were subjected to these filters), thus bringing them closer to those typically reported in high-throughput screening decks in reality and lower than those of artificially constructed data sets such as DUD [28], DUD-E [29], or DEKOIS 2.0 [30]. This not only denotes the particular challenge brought about by our data set but, also, highlights the importance of detecting and removing false positives in assembling active substances.

### 4.3. Possible Chemical Bias in Assembling Active and Inactive Substances

As previously mentioned, a noteworthy issue of raw data published on PubChem BioAssay lies in the chemically biased composition of active and inactive substances for a particular target. More specifically, there may exist “analog bias” [93] present among the molecules constituting a ligand set, which likely leads to overly good performances of virtual screening methods. This bias is generally observed in data collections whose actives (or inactives) share similar chemical features, meaning a large number of these molecules are issued from the same (or similar) scaffolds [76]. As ligand-based and structure-based screening methods tend to recognize compounds of the same chemical series, such bias may result in an overestimation of in silico screening performances [76]. Besides, significant differences between active and inactive molecules, in terms of physicochemical properties, such as molecular mass, octanol-water partition coefficient, or atomic formal charge, may as well be the source of artificial enrichment [80]. Raw experimental data from PubChem BioAssay therefore need to be finely tuned before further use, by filtering out most compounds representing the same scaffold while ensuring that the physicochemical parameters of all included molecules are kept within the same range, so that the chemical bias, if there were any, in the ligand set would be reduced [76]. An example of the importance of filtering the input data can be seen in the MTORC1 ligand set (Figure 5) included in our recently introduced LIT-PCBA data collection [22], comprising the molecules tested for an inhibitory activity towards the mTORC1 signaling pathway, targeting the human serine/threonine-protein kinase mTOR.

As to be expected, the full PubChem BioAssay data feature a larger number of scaffold clusters, with 59 clusters for the active set and 1151 clusters for the inactive set (against 41 and 1106 clusters in the LIT-PCBA active and inactive ligand sets, respectively). However, only 18 (out of 342; 5.26%) true actives possess unique scaffolds, meaning nearly 95% of all active substances in the full PubChem ligand set share chemical similarities with at least another active. Notably, nine clusters are reported to have more than 10 representatives (Figure 5A and Appendix A). The pruned LIT-PCBA active ligand set, on the other hand, includes no cluster with over 10 members and 21 clusters (51.22%) with only one substance for each. This means nearly a quarter of the LIT-PCBA active molecules (over four times the value observed in the full PubChem set) possess unique scaffolds. Moreover, the number of ligands falling into each cluster in the filtered LIT-PCBA active set is greatly reduced in comparison to that of the unfiltered data (Figure 5A and Appendix A). On the other hand, around 25% of PubChem molecules were deemed to have extreme physicochemical properties and were therefore discarded as the MTORC1 ligand set was constructed [22]. These observations suggest that (i) there is indeed significant chemical bias in the full PubChem active ligand composition, and (ii) the filtering steps that were applied to build the LIT-PCBA data collection helped reduce this bias by lowering the number of active substances sharing the same chemical features (thus avoiding the presence of too many molecules issued from the same chemotype) and by ruling out compounds that were too different from others (hence, preventing artificial enrichment). A similar conclusion can be drawn from the full PubChem inactive ligand set and the corresponding LIT-PCBA data (Figure 5B and Appendix A). The benefit of filtering the PubChem ligands in reducing the chemical bias is again highlighted as the data sets undergo a subsequent unbiasing procedure using the previously described asymmetric validation embedding (AVE) method [73], which measures pairwise distances in the chemical space between molecules belonging to four sets of compounds (training actives, training inactives, validation actives, and validation inactives; training-to-validation ratio = 3) based on the ECFP4 fingerprints. A nearly zero overall bias value (0.001) was obtained from the LIT-PCBA MTORC1 ligand set after only seven iteration steps of the AVE genetic algorithm (GA) [22], while 16 GA iterations were necessary to bring the overall bias of the full PubChem data set down to 0.006. This denotes that the pruned LIT-PCBA ligands are much less biased, in terms of chemical features, than the complete PubChem molecules and confirms the necessity of detecting chemical bias in the composition of data deposited on PubChem BioAssay and removing them, if there were any, so that the data set is better adapted for further use.

The impact of filtering the PubChem BioAssay molecules on the subsequent retrospective screening performances can also be observed with the use of two in silico methods: a 2D similarity search using extended-connectivity ECFP4 fingerprints with Pipeline Pilot [95,97] (ligand-based) and molecular docking with Surflex-Dock (structure-based) [98]. Both data sets (the full PubChem data and the pruned LIT-PCBA MTORC1 ligands) underwent the same screening protocols using the two aforementioned programs, as described in our previous paper [22]. The screening performance was evaluated according to the EF1% (enrichment in true actives at a constant 1% false positive rate over random picking) values obtained by the “max-pooling approach”, taking into account all available PDB templates of the protein target (*n* = 11), while generating only one hit list that facilitated the post-screening assessments [22]. It was observed that both methods performed better on the full PubChem data than on the filtered LIT-PCBA ligand set (Table 2). Interestingly, the true actives that were retrieved along with the top 1% false positives belonged to the same scaffold clusters or to clusters that were similar to each other. Such observations reconfirm that (i) ligand-based and structure-based screening approaches tend to recognize compounds that share chemical features, and (ii) the chemical bias present in the complete PubChem data indeed leads to overoptimistic screening performances. This, again, highlights the importance of filtering the ensemble of molecules deposited on PubChem BioAssay prior to evaluating the virtual screening procedures—first, to reduce chemical bias in the composition of the data and, then, to avoid overestimating the real discriminatory accuracy of in silico methods.

### 4.4. Potency Bias in the Composition of Active Ligand Sets

As of 30 April 2020, there were 1,067,719 small-molecule assays deposited on the PubChem BioAssay database, but only 240,999 of them (22.6%) yielded active substances with confirmed potency values. These values are provided in different terms (EC_50_, IC_50_, K_d_, and K_i_), and the threshold to distinguish true actives from true inactives varies from assay to assay, depending on the researchers who conducted the experiments. Some assays accept active substances with potency values above 100 µM (e.g., AIDs 1030, 1490, and 504847), even at the millimolar level (e.g., AIDs 1045 and 1047), while, in some others, several substances with even submicromolar potency are not deemed actives (e.g., AIDs 1221, 1224, and 1345010). It is therefore comprehensible that the potency range of true actives, as well as its distribution, is quite diverse across all assays of PubChem. As active molecules with high potency towards a biological target are easier to be picked by both ligand-based and structure-based virtual screening methods [22], ligand sets with too many actives whose potency values are in the submicromolar range are prone to overestimating the real accuracy of in silico screening. PubChem BioAssay data sets, especially those composed of highly potent true actives (potency below 1 µM), need to be filtered so that the so-called “potency bias” in the composition of their active ligand sets is reduced before further use.

An illustration of this point can be taken from the LIT-PCBA PPARG ligand set (27 true actives and 5211 true inactives) and the corresponding full PubChem BioAssay data (AID 743094; 78 true actives and 8532 true inactives) comprising small molecules that were tested for an agonistic activity on the peroxisome proliferator-activated receptor gamma (PPARg) signaling pathway [22]. The number of true actives with high potency (EC_50_ < 1 µM) in the complete PubChem data is 19, nearly three times higher than that of the pruned LIT-PCBA ligand set (*n* = 7). Upon carrying out a 2D similarity search with Pipeline Pilot using ECFP4 fingerprints and ten structurally diverse crystallographic PPARg agonists randomly chosen from 138 available structures on the Protein Data Bank as templates, it was observed that, as expected, the screening protocol managed to retrieve more highly potent true actives from the full data set than from the filtered ligand set in 70% of the cases (Figure 6). Moreover, the “max-pooling” approach, when applied to the complete PubChem data, selected seven highly potent actives among the top 1% ranked molecules, seven times higher than the amount obtained from LIT-PCBA. Among them, four even had potency values below 0.1 µM. The same screening method, on the other hand, failed to retrieve any true actives with EC_50_ < 0.1 µM from the pruned PPARG data. The screening performance observed on the full ligand set was, as a result, better than that obtained after ligand-filtering, as the EF1% value was nearly twice higher than that received with LIT-PCBA ligands. This reconfirms that in silico screening procedures tend to recognize molecules with high potency towards a protein target, and the presence of too many highly potent ligands in the data likely leads to a better screening performance. It is therefore recommended that one should filter the ensemble of PubChem BioAssay ligands to ensure that there are not too many true actives with high potency that remain, in order to avoid possible “potency bias” in the data set and the subsequent overestimation of in silico methods’ discriminatory power.

### 4.5. Processing Input Structures Prior to Virtual Screening

PubChem BioAssay ligands, as deposited on the database, can be downloaded either as SMILES strings [99] or in 2D SDF (spatial data file) format [100] and are therefore, in general, not yet ready to be directly employed as the input for most in silico screening protocols (except for 1D or 2D ligand-based approaches). A rigorous ligand-processing procedure is thus necessary to afford ready-to-use structures for virtual screening. This process concerns a wide range of aspects inherent in the three-dimensional structural formula of a molecule, including atomic coordinates in the 3D space, a formal charge assigned on each atom, the presence of different protonation states and tautomeric shifts that slightly alter the structure, and the representation of undefined stereocenters or flexible rings, as well as the existence of multiple conformations and/or configurations [101]. Various studies have concluded that database-processing has indeed an impact on the screening performance; some processing stages are even indispensable to certain programs [101,102,103,104]. Kellenberger et al. (2004) [103], Perola and Charifson (2004) [104], and Cummings et al. (2007) [101] pointed out that the initial conformation and orientation in the 3D space of a molecule, which are determined based on details featured in the original SMILES string, may significantly affect the final enrichment output by a docking program. The performances of structure-based screening methods whose scoring functions rely on ligand-receptor interactions [105,106] may be sensitive to a change in the explicit hydrogen assignment or protonation states, as the positions of hydrogen-bonding groups and proton-carrying atoms are crucial to properly detecting intermolecular hydrogen bonds and ionic interactions, respectively [101,107]. While a generation of correct multiple conformers for a molecule is not imperative when it comes to carrying out docking with GOLD [108] or Surflex-Dock [98], this step has, in fact, a pivotal role in the 3D shape similarity search using ROCS (OpenEye) [109]. The examples mentioned above denote that good in silico screening outcomes do require the careful treatment of input ligand sets, and a thorough investigation of different data-processing procedures with commonly used programs (e.g., Protoss [110], Corina [111], MOE [112], Sybyl [113], and Daylight [114]) is thus recommended. If it is possible (if the data size is not too large), one should check each output structure by hand to ensure that the assigned atom types, bond types, stereochemical properties, and protonation states are correct before further use. This also applies to the protein structure preparation prior to screening, as structural features of the protein target, especially those of the binding site, are of indisputable importance to the structure-based virtual screening performance.

## 5. Conclusions

Retrieving experimental PubChem BioAssay data to construct novel data sets for virtual screening evaluations helps avoid assuming false negatives among inactive ligands, which is a problem inherent in artificially developed data collections. However, there remain several issues regarding assay selection, false active molecules, chemical bias, and potency bias, as well as data curation, which are worth noticing prior to employing PubChem input for database-designing purposes. To the best of our knowledge, there have been several publicly available data sets that were constructed from the data deposited on this repository, but the quantity is not yet considerable, and there still exist some limitations in the design of these data collections. More efforts in this regard are recommended, with the points raised in this manuscript taken into account, in order to offer more realistic data sets suitable for validating both ligand-based and structure-based in silico screening procedures in the future. Of course, the herein proposed good practices should also be applied to proprietary bioactivity data.

## Figures and Tables

**Figure 1 ijms-21-04380-f001:**
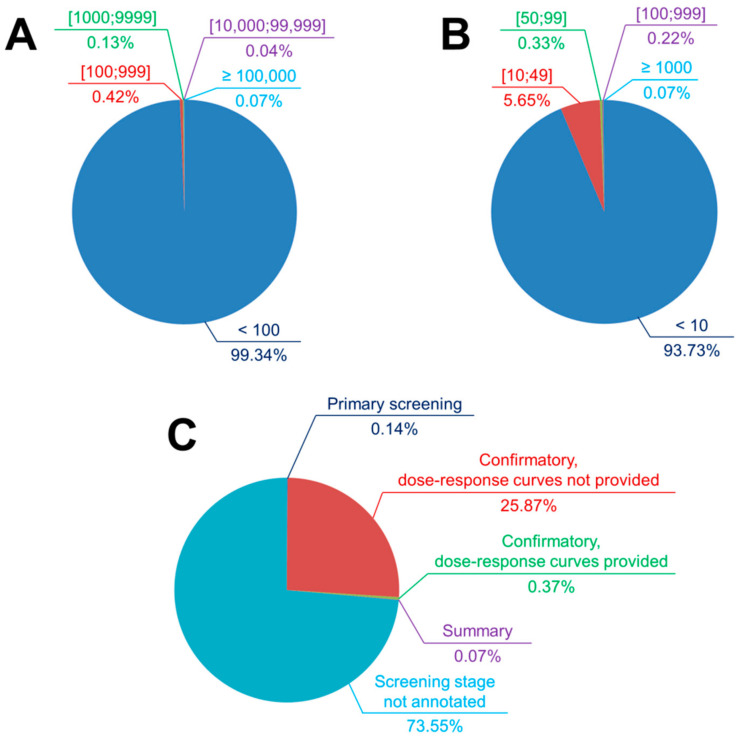
Partition of small-molecule PubChem bioactivity assays according to the number of tested substances (**A**), the number of active substances (**B**), and the screening stage (**C**). It is observed that most assays are small-scale screening projects in which fewer than 100 substances were tested and no more than nine actives were identified. All statistics were updated as of 30 April 2020.

**Figure 2 ijms-21-04380-f002:**
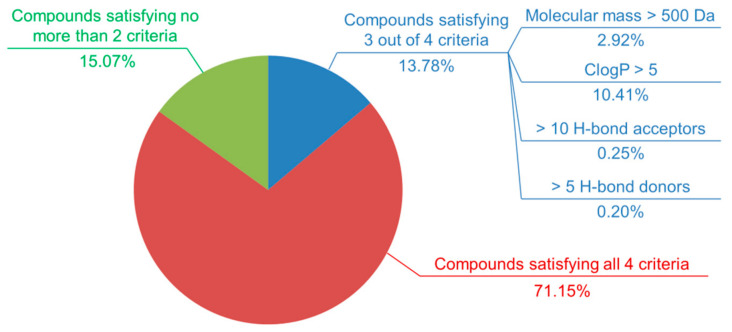
Partition of compounds tested in PubChem bioactivity assays according to four criteria of the Lipinski’s rule of five. It is observed that most compounds (over 70%) satisfy all criteria. Nearly 85% of deposited compounds violate no more than one criterion. On the other hand, only 0.1% of all compounds (over 130,000) do not satisfy any criterion. Statistics were updated as of 30 April 2020.

**Figure 3 ijms-21-04380-f003:**
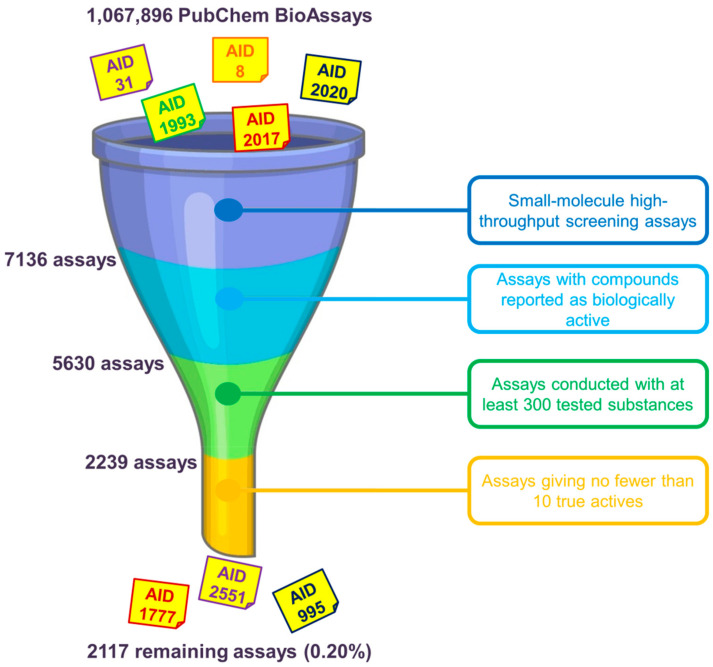
Primary selection of PubChem assays whose ligand sets should be further considered for evaluating virtual screening methods. We herewith recommend the use of only small-molecule high-throughput screening (HTS) assays giving at least 10 biologically active molecules among no fewer than 300 tested substances. Overall, there are only 2117 assays (0.20% of 1,067,896 assays in total, as of 30 April 2020) that remain, indicating a very small portion of PubChem assays that may be considered after this initial check.

**Figure 4 ijms-21-04380-f004:**
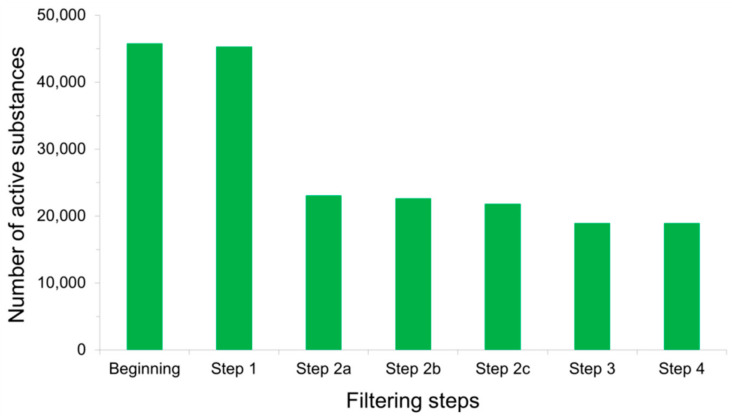
Total number of active substances that remained after each filtering step was applied to PubChem BioAssay ligands during the construction of the LIT-PCBA data set [22]: Step 1—inorganic molecules; Step 2a—actives with Hill slopes <0.5 or >2; Step 2b—actives with a frequency of hits >0.26; Step 2c—actives found among 10,892 confirmed aggregators, luciferase inhibitors, or auto-fluorescent molecules; Step 3—substances with extreme molecular properties; and Step 4—3D conversion and ionization failures. It can be observed that the sole step 2a removed the most active molecules (over 50% of them), thus significantly reducing the population of true actives in comparison to that of true inactives.

**Figure 5 ijms-21-04380-f005:**
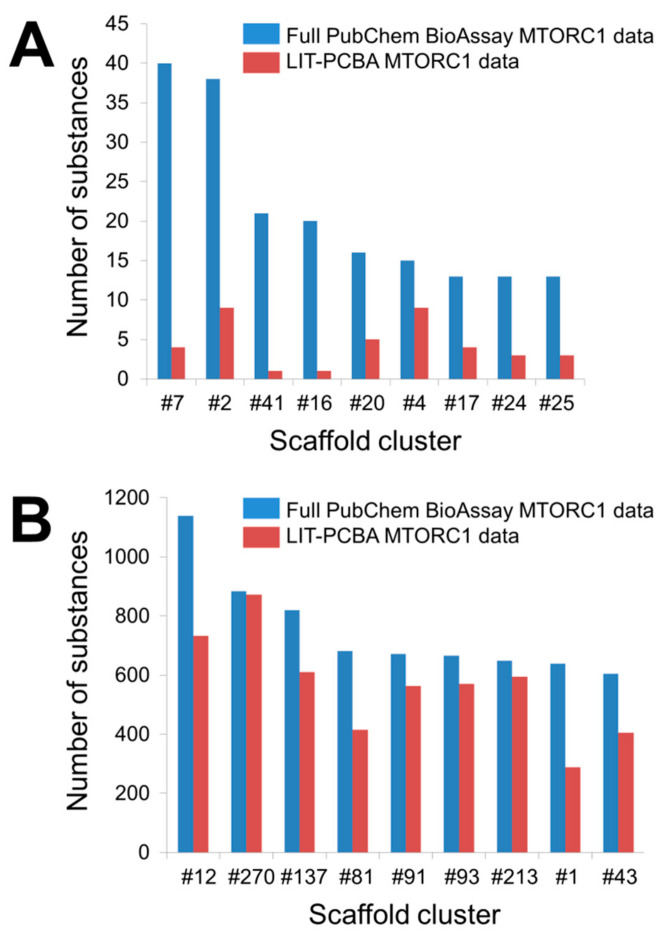
Number of substances falling into each scaffold cluster that includes more than 10 true active molecules (**A**) or 600 true inactive molecules (**B**). Bemis-Murcko frameworks derived from the input molecules were first created by trimming each active and each inactive separately with Pipeline Pilot 19.1.0.1964 [94,95]. A hierarchical scaffold tree consisting of canonical SMILES (simplified molecular-input line-entry system) strings that represent the rings, linkers, and double bonds in each molecule was next generated according to an iterative ring-trimming procedure described by Schuffenhauer et al. (2007) [96]. All ligands were then clustered based on the smallest scaffold at the root of the scaffold tree for each ligand. The number that follows each hash symbol indicated in this figure refers to the ordinal number of a scaffold cluster as issued by Pipeline Pilot. Details of all clusters can be found in the Appendix A.

**Figure 6 ijms-21-04380-f006:**
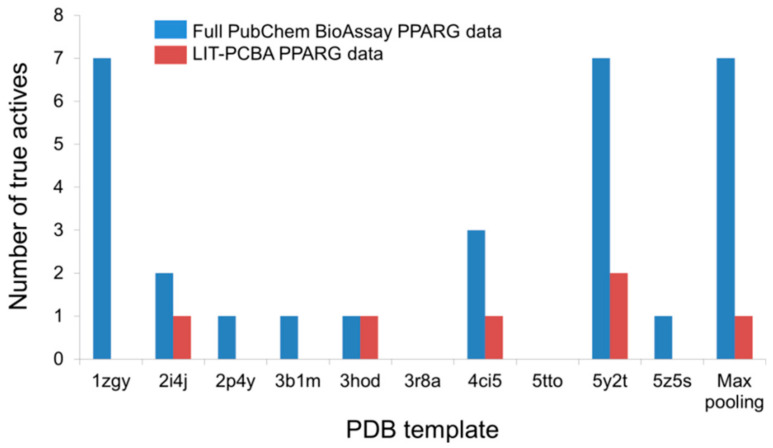
The number of highly potent true actives (EC_50_ < 1 µM) retrieved among the top 1% ranked molecules by a 2D ECFP4 fingerprint similarity search from the full PubChem BioAssay data and the corresponding LIT-PCBA PPARG ligand set after ligand-filtering. Ten known crystallographic PPARg agonists were randomly chosen as templates from 138 available structures on the Protein Data Bank.

**Table 1 ijms-21-04380-t001:** Overview of the main open-access benchmarking data sets developed from experimental PubChem BioAssay data.

Data Sets	Year	Number of Ligand Sets	Number of Molecules Per Ligand Set	Active-to-Inactive Ratio	Assay Data		Assay Artifacts Avoided	Chemical Bias Avoided	Virtual Screening Suitability	
					Primary	Confirmatory			Ligand-Based	Structure-Based
MUV [80]	2009	17	15,030	2 × 10^−3^	✓	✓	✓	✓	✓ ^a^	✓

UCI [81]	2009	21	69 to 59,795	2 × 10^−4^ to 0.33	✓	✓			✓	

Butkiewicz et al. [82]	2013	9	61,849 to 344,769	5 × 10^−4^ to 7 × 10^−3^		✓			✓	

Lindh et al. [83]	2015	7	59,462 to 338,003	7 × 10^−5^ to 1 × 10^−3^	✓	✓	✓	✓	✓	✓

LIT-PCBA [22]	2020	15	4247 to 362,088	5 × 10^−5^ to 0.05		✓	✓	✓ ^b^	✓	✓

^a^ Ligand-based approaches are preferred. ^b^ Unbiased training and validation sets are provided for machine learning. MUV: maximum unbiased validation.

**Table 2 ijms-21-04380-t002:** Retrospective screening performance of a 2D ECFP4 fingerprint similarity search with Pipeline Pilot and molecular docking with Surflex-Dock on the full PubChem BioAssay data and the pruned LIT-PCBA MTORC1 ligand set, demonstrated by the enrichment in true actives at a constant 1% false positive rate over random picking (EF1%) values and the numbers of true actives retrieved along with the top 1% false positives by the “max-pooling” approach.

Data Sets	2D ECFP4 Fingerprint Similarity Search		Molecular Docking	
	EF1%	Number of Retrieved Actives	EF1%	Number of Retrieved Actives
Full PubChem data	0.6	2	3.2	11
LIT-PCBA MTORC1 data	0.0	0	1.0	1

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
