# Peer review of "Benchmarking Data Sets from PubChem BioAssay Data: Current Scenario and Room for Improvement"

_ijms, 2020, doi:10.3390/ijms21124380_

Round 1
Reviewer 1 Report
The submitted manuscript entitled “Benchmarking Data Sets From PubChem BioAssay Data: Current Scenario and Room for Improvement” by Viet-Khoa Tran-Nguyen and Didier Rognan provides the comprehensive overview of data collection retrieved from the experimental PubChem BioAssay database that can be potentially useful in the virtual ligand/structure-based screening protocols. Moreover, the presented benchmarking dataset was compared with the artificially generated DUD or DEKOIS ensembles, respectively.
In fact, the reviewed paper might be interesting for the scientific community, because it contributes considerably to the medicinal chemistry field. Basically, the work is well organized and written; however some minor concerns should be addressed before publishing the manuscript in IJMS.
Let me show some points below.
- Line 87-88 and 274. Authors stated that ‘over 95% of which are organic molecules’ and ‘a series of consecutive filters, which ruled out inorganic chemicals’. There is no clear boundaries between organic and inorganic compounds, for instance carbonyl dichloride (ClCOCl) and urea (NH2CONH2). What kind of filter was applied in this case? Please, provide short description in the text of the manuscript.
- Line 329. It is stated that ‘the selection of only small-329 molecule HTS assays yielding biologically active molecules’. What criteria were taken into consideration to single out small molecules. Please, provide short description.
- Line 465. The word ‘expected’ is doubled, probably by mistake.
- Figure 5. What does mark ‘#’ mean in Figure A and B?
Author Response
We thank the reviewer for these nice comments. His comments on the manuscript have been taken into account as follows:
- Line 87-88 and 274. Authors stated that ‘over 95% of which are organic molecules’ and ‘a series of consecutive filters, which ruled out inorganic chemicals’. There is no clear boundaries between organic and inorganic compounds, for instance carbonyl dichloride (ClCOCl) and urea (NH2CONH2). What kind of filter was applied in this case? Please, provide short description in the text of the manuscript.
Response: The term “organic molecules” used in this manuscript refers to molecules that bear no atom other than H, C, N, O, P, S, F, Cl, Br, and I. This definition was also employed during the construction of our LIT-PCBA data set. The authors have added a short description in this regard in the revised manuscript (marked in red, section 2 – page 3, section 3.5 – page 8).
- Line 329. It is stated that ‘the selection of only small-molecule HTS assays yielding biologically active molecules’. What criteria were taken into consideration to single out small molecules. Please, provide short description.
Response: As stated in the manuscript, in the PubChem BioAssay database, there are two types of assays: small-molecule assays and RNA interference ones. RNAi assays were conducted on microRNA-like molecules comprising twenties of base pairs that violate most drug-likeness rules of thumb, and are therefore, not of great interest in drug discovery. That is why we recommend the use of only small-molecule assays. The authors have added an explanation (marked in red) in section 4.1.1, page 9 in the revised manuscript.
- Line 465. The word ‘expected’ is doubled, probably by mistake.
Response: We thank the reviewer for this remark. The second “expected” has been removed in the revised manuscript (page 13).
- Figure 5. What does mark ‘#’ mean in Figure A and B?
Response: The number that follows each hash symbol (#) indicated in this figure refers to the ordinal number of a scaffold cluster as issued by Pipeline Pilot. The authors have added an explanation in the caption of Figure 5 (marked in red) in the revised manuscript.
Reviewer 2 Report
This is an excellent review of the Benchmarking Data Sets From PubChem BioAssay data. The authors reviewed the history, the challenges, solutions to address the issues and also provide their visions on the future directions. We know that in the in silico screening process, scientists have been struggling on finding a good standardized dataset to test their approaches, and I believe this paper can answer most of the concerns we might have. Good job.
Author Response
Response: The authors would like to thank the reviewer for these encouraging comments.
Reviewer 3 Report
Authors have reviewed data collections from PubChem BioAssay and a comprehensive discussion associated issues to consider while designing ligand sets. This is potentially informative for both the informatics community and ligand/structure-based drug discovery groups. Presented a snapshot of history and various reasons for failure of several data sets and the importance of target tested collection of compounds from databases that include Bioactivities.
The review is lengthy and some information can be moved to the supplementary section - as in Tables 1,2 and Figures 1 and 2 - this information can succinctly summarized in the text. Consider sub-sections of lengthy paragraphs.
Author Response
The review is lengthy and some information can be moved to the supplementary section - as in Tables 1,2 and Figures 1 and 2 - this information can succinctly summarized in the text. Consider sub-sections of lengthy paragraphs.
Response: The authors have moved the original Tables 1 and 2 to the Supplementary Materials, now as Tables S1 and S2. The ordinal numbers of the other tables have been modified accordingly in the revised manuscript. Moreover, the authors have divided the section 3 into five subsections (from 3.1 to 3.5), and the section 4.1 into three smaller sub-sections (from 4.1.1 to 4.1.3) in the revised manuscript.